# Removal of Phosphate from Aqueous Solution by Zeolite-Biochar Composite: Adsorption Performance and Regulation Mechanism

**Zhaoxia Deng** [1,2], **Shangyi Gu** [1], **Hongguang Cheng** [2,*], **Dan Xing** [3], **Gratien Twagirayezu** [4], **Xi Wang** [2,5], **Wenjing Ning** [2,6] **and Mingming Mao** [1,3]

1    College of Resources and Environmental Engineering, Guizhou University, Guiyang 550025, China; dengzhaoxia@mail.gyig.ac.cn (Z.D.); sygu@gzu.edu.cn (S.G.); mminggo@163.com (M.M.)
2    State Key Laboratory of Environmental Geochemistry, Institute of Geochemistry, Chinese Academy of Sciences, Guiyang 550002, China; wanggoforit@webmail.hzau.edu.cn (X.W.); ningwenjing@mail.gyig.ac.cn (W.N.)
3    Guizhou Academy of Agricultural Science, Institute of Pepper Guiyang, Guiyang 550000, China; 2004xingdan@163.com
4    School of Environmental and Municipal Engineering, Lanzhou Jiaotong University, Lanzhou 730070, China; tgratien0@gmail.com
5    Key Laboratory of Arable Land Conservation (Middle and Lower Reaches of the Yangtze River), Ministry of Agriculture, College of Resource and Environment, Huazhong Agricultural University, Wuhan 430070, China
6    College of Resources and Environment, Yangtze University, Wuhan 430100, China
*    Correspondence: chenghongguang@vip.gyig.ac.cn

**Abstract:** Recently, rampant eutrophication induced by phosphorus enrichment in water has been attracting attention worldwide. However, the mechanisms by which phosphate can be eliminated from the aqueous environment remain unclear. This study was aimed at investigating the adsorption performance and regulation mechanisms of the zeolite-biochar composite for removing phosphate from an aqueous environment. To do this, physicochemical properties of the zeolite-biochar composite were assessed by Fourier transform infrared spectroscopy (FTIR), scanning electron microscopy (SEM), Brunauer–Emmett–Teller (BET) specific surface area (SSA) analyzer, and transmission electron microscopy (TEM). Adsorption tests were performed to evaluate the adsorption ability of the composite material for mitigating excess phosphorus in the aqueous environment. The findings evinced that the phosphorus removed by PZC 7:3 (pyrolyzed zeolite and corn straw at a mass ratio of 7:3) can reach 90% of that removed by biochar. The maximum adsorption capacities of zeolite, biochar, and PZC 7:3 were 0.69, 3.60, and 2.41 mg/g, respectively. The main mechanism of phosphate removal by PZC 7:3 was the formation of thin-film amorphous calcium-magnesium phosphate compounds through ligand exchange. This study suggests that PZC 7:3 is a viable adsorbent for the removal of phosphate from aquatic systems.

**Keywords:** phosphate; natural zeolite; biochar; composite; co-pyrolysis

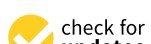



## 1. Introduction

As an essential nutrient element, phosphorus supports the growth and metabolism of living organisms, and its content and supply capacity directly regulate the production level of the plant [1]. Recently, large quantities of phosphorus have been mined from mineral deposits and used as fertilizer in agricultural activities to satisfy the demand for food [2]. Commonly, a high amount of fertilizer input ensures crop production yield in the soil [3]. However, the low utilization efficiency of fertilizer results in a considerable quantity of nutrients remaining in the soil [4], especially phosphorus, for which the utilization rate is only 10–15% [5]. On the one hand, phosphorus enters the water body with rainfall and

water flow scouring, leading to water pollution and even eutrophication [6,7]. Several studies have revealed that high concentrations of phosphorus stimulate the uncontrolled growth of algae and excessive consumption of dissolved oxygen, resulting in the extinction of fish and shrimp and the deterioration of water quality, thus affecting human health [8,9]. On the other hand, it was evinced that the large-scale exploitation of phosphorus resources will eventually lead to a shortage [10]. Therefore, removing phosphorus from water is highly needed to prevent and control water pollution [11].

There are several technologies that can be utilized for removing phosphate from water, including physiochemical technologies and biological methods [12–15]. These methods are costly, operate under sensitive conditions, and do not recover phosphorus resources [16,17]. Presently, the adsorption method has become a common and effective water treatment method for removing phosphate because it is easy to use, cheap, and convenient [8]. In addition, after adsorbing phosphorus, the adsorbent can be used again as a phosphorus resource [18,19]. However, the selection of the adsorbent is crucial for phosphorus removal by adsorption [20]. Therefore, it is vital to make a cheap and easy-to-operate adsorbent in order to use the adsorption methods universally.

Natural minerals such as zeolite, montmorillonite, and diatomite have been widely used as adsorbents to improve the adsorption of phosphate [21–23]. Among them, zeolite is considered as a wastewater treatment material with a wide application prospect due to its abundant resources, low price, and stable structure [24]. However, due to the negative charge resulting from the homogeneous substitution of cations in the crystal lattice, zeolite has excellent adsorption for removing cation pollutants from water. In contrast, it has low affinities for anions such as phosphorus [25]. Numerous studies reported that phosphate adsorption by zeolite was enhanced by its thermal pyrolysis and acid modification [26]. This means that after thermal pyrolysis and acid modification, the zeolite significantly enhanced the adsorption of phosphorus in water [27,28].

Biochar, a multifunctional product made from agricultural waste or household waste, has gained popularity worldwide [29,30]. In addition to the improvement of agricultural soil, enhancement of agricultural production, and carbon sequestration [31], biochar has been widely utilized in the remediation of soil polluted by heavy metals [32] and water pollution control [33]. However, the implications of biochar application on water pollution, such as phosphorus removal, are regulated by the variation of biochar physicochemical properties, which result from the diverse feedstock and pyrolysis regimes [34,35]. The main obstacles to the widespread use of biochar to remove phosphorus from water are that biochar has a low density, large volume, and inconvenient transportation. In addition, biochar lacks attachments and cannot be easily separated from the liquid, which also hinders its application [36].

After considering the advantages and disadvantages of zeolite and biochar for removing phosphorus from water, this study assumes that the employment of zeolite and biochar as a matrix for the preparation of zeolite-biochar composites has tremendous potential for removing phosphorus from water. The preparation process of biochar pyrolysis can indirectly provide thermal modification for zeolite, and at the same time, the organic acids released during the pyrolysis of biochar feedstock have the potential to modify zeolite for acid modification. Zeolite can provide attachments for biochar, reduce the volume of biochar, and facilitate transportation [37]. Thus, the processes for preparing zeolite-biochar composite materials may significantly lower the amount of biochar. Although the adsorption effects of the composite material on phosphorus in water are well-known, the mechanisms of its adsorption are still unclear.

Therefore, determining the optimal ratio and selecting a suitable method for the preparation of the biochar and zeolite composite material, as well as studying its adsorption performance and mechanism, is very important for reducing water eutrophication. The main intentions of this work are to: (1) prepare a zeolite-biochar composite material with the most suitable adsorption performance (proportion, mixing method), (2) explore the

influences of dosage and pH on the phosphate adsorption, and (3) highlight the phosphate adsorption mechanisms of the zeolite-biochar composite.

## 2. Materials and Methods

### 2.1. Materials

Zeolite was bought from Xinyang City, Henan Province, China. Its particle size is 150 μm, and its cation exchange capacity is 53.15 cmol/kg. The mineralogical composition of zeolite was clinoptilolite (66.3%), montmorillonite (29.3%), and quartz (4.4%). Figure 1 illustrates the XRD pattern of the material.

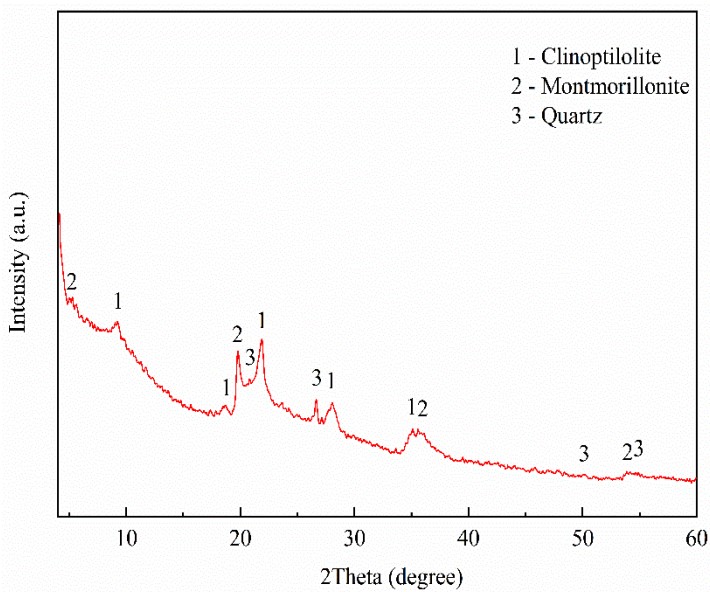

**Figure 1.** The XRD pattern of the zeolite.

Corn straw was collected from farmland in Niudachang, Longli County, Qiannan Buyi, and Miao Autonomous Prefecture, Guizhou Province, China (26°23′2″ N, 106°45′41″ E). After collection, corn straw was dried naturally and crushed for subsequent use.

The reagents used in these experiments, such as potassium dihydrogen phosphate ($KH_2PO_4$), ammonium molybdate (($NH_4$)$_2MoO_4$), and ascorbic acid, were purchased from Tianjin Kemeiou Chemical Reagent Co. Ltd. (Tianjing, China). Sulfuric acid ($H_2SO_4$) was bought from Sinopharm Chemical Reagent Co., Ltd. (Shanghai, China). All reagents were purely graded. Deionized water was used to reach the corresponding concentration when needed.

### 2.2. Preparation Processes of Zeolite-Biochar Composites

#### 2.2.1. Biochar

The corn straw was pyrolyzed at 350 °C in a tube furnace (SG-GL 1200 K, Shanghai Institute of Optimal Precision Instruments, CAS, Shanghai, China). The heating gradient was 11 °C/min and the holding time was 2 h at the peak temperature. In the whole pyrolysis, 0.15 L/min of nitrogen was used as a protective gas for isolating oxygen. For the next experiment, the product from the pyrolysis was cooled and then put through a 100-mesh sieve.

#### 2.2.2. Pyrolysis of Zeolite

The above-prepared zeolite was pyrolyzed in the tube furnace with the same process. In this study, it was named PZ.

### 2.2.3. Pyrolysis of Zeolite and Biochar

Firstly, zeolite was pyrolyzed with the same procedure as biochar, and then pyrolyzed zeolite was mixed with biochar. The mixed material was named "pyrolyzed zeolite and biochar". Similarly, the different mass ratios of pyrolyzed zeolite and biochar material were produced as mentioned above, where the proportions were 9:1, 8:2, and 7:3 for materials named PZB 9:1, PZB 8:2, and PZB 7:3, respectively.

### 2.2.4. Pyrolysis of Zeolite and Corn Straw

Under the same processes as with biochar, zeolite and corn straw were mixed and then pyrolyzed in the tube furnace. As for the different ratio materials, based on the yielded biochar and pyrolyzed zeolite, different amounts of corn straw and zeolite were mixed and then pyrolyzed. The mass ratios of zeolite to biochar in different products were 9:1, 8:2, and 7:3 for materials named PZC 9:1, PZC 8:2, and PZC 7:3, respectively.

### 2.3. Characterization of Zeolite-Biochar Composites and Phosphate Determination

At a mass ratio of adsorbent to deionized water of 1:10 (g/mL), electrical conductivity and pH were measured using a conductivity meter (DDS-307, Shanghai Leici Instrument Co., Ltd., Shanghai, China) and a pH meter (pHS-3E, Shanghai Leici Instrument Co., Ltd.). X-ray diffraction (XRD) analysis was carried out on a D/MAX2000 diffractometer (Rigaku Co., Tokyo, Japan). SSA was determined by the BET method with a surface area analyzer (ASAP 2460, Micromeritics, Norcross, GA, USA). SEM and TEM with an energy dispersive spectrometer analyzer (JSM-IT 300, Japan electronics Co., Ltd., Tokyo, Japan; Tecnai $G^2$ F20 S-Twin, FEI Inc., Valley City, ND, USA) were used to describe the surface morphology and elemental composition of the materials. An FTIR spectrometer was used to characterize the functional groups on the adsorbent surface in the spectrum region of 400 to 4000 cm$^{-1}$ (Nicolet iS 5, Thermo Scientific, Waltham, MA, USA). An elemental analyzer (Elementar Varioel Cube, Hanau, Germany) was used for elemental analysis. The Malvern Zetasizer Nano ZSE (Malvern Instruments, Malvern, UK) was used to measure zeta potentials. The phosphate concentration in the filtrate was measured based on the colorimetric molybdate blue method (Murphy and Riley 1962).

### 2.4. Adsorption Experiment

### 2.4.1. Phosphate Removal Efficiency

Optimal dosage experiment: Zeolite, biochar, and PZC 7:3 were chosen to find the optimal dosage for phosphate removal. Firstly, 0.2, 0.4, 0.6, 0.8, 1.0, and 1.2 g of the three materials were weighed and put into 50 mL polyethylene (PE) centrifuge tubes, respectively. After that, 20 mL of a 50 mg/L phosphate solution was added into the tube. Then, PE centrifuge tubes were positioned in a constant temperature oscillator at 25 °C and 200 rpm for 240 min. Then, PE centrifuge tubes were placed in a centrifuge (4000 rpm) for 10 min, followed by filtration of the supernatant with filter paper. The obtained supernatant was finally analyzed to determine the phosphate concentration in the solution. The optimal dosage of 0.8 g was selected for the next experiment based on the above experiment. In this study, every treatment was carried out four times to ensure accurate results.

Optimal mixed-material adsorption experiments: Nine materials (zeolite, PZ, biochar, PZB 9:1, PZB 8:2, PZB 7:3, PZC 9:1, PZC 8:2, and PZC 7:3) were used to carry out the phosphate removal experiment. Firstly, 0.8 g of the above material was put into a 50 mL centrifuge tube, followed by adding 20 mL of a 50 mg/L phosphate solution. After the same procedure as in the above experiment, phosphate in the solution was measured. According to the results of this experiment, it was shown that PZC 7:3 had the highest phosphate removal rate compared to the biochar and zeolite composite. The next experiment was performed with zeolite, biochar, and PZC 7:3.

### 2.4.2. The Influence of pH on the Adsorption of Phosphate Adsorption

To determine the influence of pH on the adsorption performance of zeolite, biochar, and PZC 7:3, a solution of 0.1 mol/L HCl and NaOH was used for adjusting the pH of the solution to 3.0, 5.0, 7.0, 9.0, and 11.0. Thereafter, 0.8 g of zeolite, biochar, and PZC 7:3 was poured into a 50 mL centrifuge tube, followed by 20 mL of a 50 mg/L phosphate solution with a different pH. All the above experiments were placed in PE centrifuge tubes in a constant temperature oscillator at 25 °C and 200 rpm for 240 min. Then, the centrifuge (4000 rpm) was avaliable to measure the samples for 10 min, of which the supernatant was applied so as to determine the phosphate concentration in the solution.

### 2.4.3. Adsorption Kinetics and Isotherms

Adsorption kinetics was investigated by adding 0.8 g of zeolite, biochar, and PZC 7:3 into a 50 mL PE centrifuge tube that contained 20 mL of a 50 mg/L phosphate solution at pH = 7. Then, the centrifuge tube was placed in a constant temperature oscillator set to 25 °C and 200 rpm and oscillated for 5, 10, 20, 30, 60, 120, 240, 480, 960, and 1440 min. After that, the supernatant was taken and stored for analysis. The adsorption isotherms of phosphate were investigated using 0.8 g of zeolite, biochar, and PZC 7:3 in 20 mL of an initial concentration of 5, 10, 25, 50, 100, 200, and 400 mg/L phosphate solution at pH = 7. After that, the sample was circulated in a constant temperature oscillator set to 25 °C for 240 min at 200 rpm. The supernatant was determined based on the experiment mentioned above.

### 2.5. Desorption Experiment

The desorption experiment can be used as one of the indexes to judge the stability of an adsorbent. Firstly, the phosphate-loaded adsorbents were prepared by weighing 0.8 g of zeolite, biochar, and PZC 7:3 into 50 mL PE centrifuge tubes. After that, 20 mL of a 50 mg/L phosphate solution (pH = 7) was added. The mixture was kept at a constant temperature, where it was oscillated at 25 °C and 200 rpm for 4 h. Thereafter, it was centrifuged, followed by the removal of the supernatant. Then, the adsorbents were dried in an oven at 60 °C. Finally, 0.8 g of dried phosphate-loaded zeolite, biochar, and PZC 7:3 was weighed separately and then 20 mL of deionized water was added and then shaken for 24 h for desorption experiments. The samples were centrifuged and filtered, followed by the determination of the phosphate concentration based on the above-mentioned method used to calculate the desorption rate of phosphate.

### 2.6. Experimental Data Analysis

### 2.6.1. Adsorption Capacity and Removal Rate

The equilibrium adsorption capacity and removal rate of phosphate were determined based on the following formula:

$$q_e = \frac{V(C_0 - C_e)}{m} \tag{1}$$

$$R = \frac{C_0 - C_e}{C_0} \times 100\% \tag{2}$$

where $q_e$ (mg/g) denotes the equilibrium adsorption capacity, $V$ (L) represents the volume of the solution, $m$ (g) is the mass of the material, $C_0$ and $C_e$ (mg/L) represent the initial and equilibrium concentrations of phosphate, and $R$ (%) represents the percentage of phosphate removed.

### 2.6.2. Adsorption Kinetics

Pseudo-first-order kinetics and pseudo-second-order kinetical models were utilized to fit the experimental values [38].

Pseudo-first-order model:

$$\ln(q_e - q_t) = \ln q_e - k_1 t \tag{3}$$

Pseudo-second-order model:

$$\frac{t}{q_t} = \frac{1}{k_2 q_e^2} + \frac{t}{q_e} \tag{4}$$

where $q_e$ and $q_t$ (mg/g) are the phosphate concentrations at equilibrium and time $t$, and $k_1$/min and $k_2$ (g/(mg min)) are model-specific rate constants.

2.6.3. Adsorption Isotherms

Langmuir and Freundlich models were used to characterize the adsorption mechanisms and adsorption equilibrium [39].

Langmuir isotherm model:

$$\frac{C_e}{q_e} = \frac{1}{K_L q_m} + \frac{1}{q_m} C_e \tag{5}$$

Freundlich isotherm model:

$$\lg q_e = \lg K_F + \frac{1}{n} \lg C_e \tag{6}$$

where $q_m$ (mg/g) represents the Langmuir maximum adsorption capacity, $C_e$ (mg/L) represents the equilibrium solution concentration of the phosphate, and $K_L$, $K_F$, and $n$ are all constants.

All values are means $\pm$ SD ($n = 4$). Differences between treatments were examined by one-way ANOVA, and different superscript letters represent significant differences between treatments at the $p < 0.05$ level. Statistical analysis was performed using SPSS software version 19.0 and Origin 2018.

## 3. Results and Discussion

### 3.1. Physicochemical Characterization of Zeolite, Biochar, and PZC 7:3

3.1.1. Basic Physicochemical Properties

Table 1 displays the physicochemical characteristics of zeolite, biochar, and PZC 7:3, such as the pH value, electrical conductivity (EC), and hydrogen/carbon ratio (H/C). The pH levels of zeolite, biochar, and PZC 7:3 were all alkaline. As in previous reports, oxygen-containing functional groups generated during pyrolysis resulted from an increased pH [40,41]. In addition, it was shown that alkaline ions such as $Na^+$, $K^+$, $Mg^{2+}$, and $Ca^{2+}$ remaining in the biochar ash led to a high pH of biochar [42]. PZC 7:3 was a combined composite of biochar and zeolite. Its pH was increased compared to zeolite due to the influence of biomass carbonization, which is similar to the research findings of Mosa [43]. Generally, H/C can reflect the aromaticity of biochar, and a smaller H/C indicates higher aromaticity [44]. The H/C ratio of the PZC 7:3 was lower than that of biochar, demonstrating that the addition of minerals enhanced the aromaticity of biochar and increased the stability of PZC 7:3, which is similar to the findings of Li [45] and Yang [46].

The SSA of an adsorbent is a parameter that characterizes its adsorption capacity [29]. Table 2 displays the pore structure parameters of the materials. The specific surface area of zeolite was the largest. The specific surface area of zeolite after pyrolysis was not significantly different ($p > 0.05$) compared to that of PZ. Still, the pore volume and pore size were significantly increased ($p < 0.05$), which may be due to the evaporation of water molecules in the pores of zeolite by heating. The specific surface area of PZC 7:3 was not significantly different ($p > 0.05$) compared to that of the biochar. The reason may be that biochar was attached to the surface of zeolite during the pyrolysis of the composite [47], so the specific surface area of PZC 7:3 was closer to that of the biochar.

**Table 1.** Physicochemical properties of materials.

| Sample | pH | EC ($\mu$s cm$^{-1}$) | H/C | Zeta Potential (mV) |
|---|---|---|---|---|
| zeolite | $8.23 \pm 0.08$ [c] | $58.48 \pm 1.25$ [c] | - | $-29.3 \pm 0.42$ [b] |
| biochar | $10.10 \pm 0.03$ [a] | $5490.00 \pm 106.54$ [a] | $0.13 \pm 0.01$ [a] | $-35.43 \pm 0.4$ [a] |
| PZC 7:3 | $9.46 \pm 0.03$ [b] | $1690.75 \pm 3.57$ [b] | $0.09 \pm 0.00$ [b] | $-29.3 \pm 0.35$ [b] |

All data represented are means $\pm$ SD ($n$ = 4). The various superscript letters indicate statistically significant differences between treatment groups ($p < 0.05$).

**Table 2.** Pore structure parameters of materials.

| Sample | SSA (m$^2$ g$^{-1}$) | Pore Volume (cm$^3$ g$^{-1}$) | Pore Size (nm) |
|---|---|---|---|
| zeolite | $30.27 \pm 0.96$ [a] | $0.0438 \pm 0.0007$ [b] | $5.80 \pm 0.10$ [c] |
| PZ | $29.86 \pm 0.43$ [a] | $0.0499 \pm 0.0003$ [a] | $6.69 \pm 0.06$ [b] |
| biochar | $2.23 \pm 0.14$ [b] | $0.0033 \pm 0.0004$ [d] | $6.02 \pm 0.15$ [c] |
| PZC 7:3 | $2.35 \pm 0.02$ [b] | $0.0056 \pm 0.0001$ [c] | $9.47 \pm 0.33$ [a] |

All values represented are means $\pm$ SD ($n$ = 4). The various superscript letters indicate statistically significant differences between treatment groups ($p < 0.05$).

### 3.1.2. SEM-EDS Analysis

Figure 2 displays the surface morphology of zeolite, biochar, and PZC 7:3. As shown in Figure 2a, biochar exhibited honeycomb and bundle structures with smooth surfaces and no filler in the pores. Figure 2b shows that the morphology of zeolite was irregular, and the surface was uneven and rough. The zeolite particle morphology is almost unchanged in PZC 7:3 (Figure 2c), but it may be affixed to the surface of biochar or incorporated into its pores. EDS results also show that PZC 7:3 (Figure 2f) combines the high oxygen, rich silicon in zeolite, and rich carbon in the biochar. Similarly, this result is consistent with the previous observations [29,43]. During the pyrolysis process, the zeolite and biomass are fully combined, and the zeolite surface can attach to the biochar, while the zeolite can enter the pores of the biochar (Figure 2c); thus, some physical and chemical reactions occur.

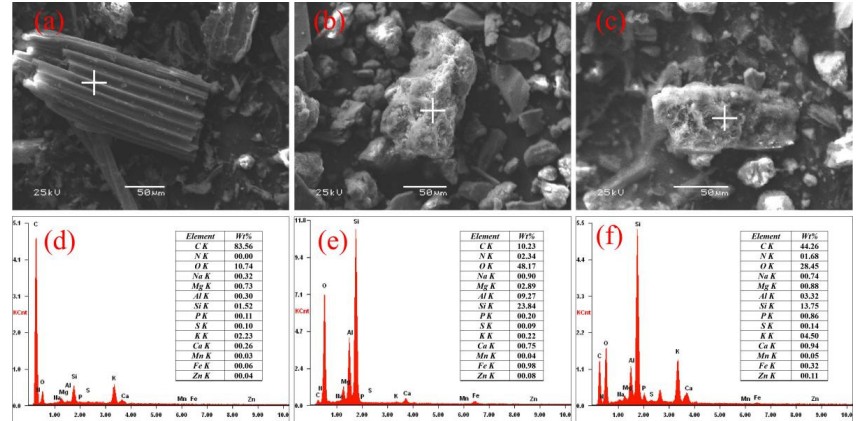

**Figure 2.** The SEM and EDS images of zeolite, biochar, and PZC 7:3 (SEM images of (**a**–**c**) biochar, zeolite, and PZC 7:3, and EDS images of (**d**–**f**) biochar, zeolite, and PZC 7:3, respectively).

### 3.1.3. FTIR Analysis

The FTIR spectra of zeolite, biochar, and PZC 7:3 before phosphate adsorption are shown in Figure 3. Compared to zeolite, the absorption peaks of 2926 and 1441 cm$^{-1}$ were found in PZC 7:3. The absorption peaks of 3630, 1040, and 519 cm$^{-1}$ were discovered in PZC 7:3 but were not seen in biochar. According to a previous publication, the stretching vibration of O-H was evinced in the range of 3628–3634 cm$^{-1}$ [48]. Both the C-H stretching

bands in the range of 2800–3000 cm$^{-1}$ and the bands in the range of 1000–1400 cm$^{-1}$ were created by the combination and overlapping of C=O stretching bands as well as numerous deformations [49,50]. In the PZC 7:3, the Si-O-Si skeleton vibration at 1040 and 519 cm$^{-1}$ represented the Si-O or Al-O bending vibration [18,51]. PZC 7:3 not only retained the stabilizing properties of zeolite, but also combined the oxygen-containing functional groups such as -COOH and -OH of biochar, indicating that PZC 7:3 might be beneficial for phosphate removal.

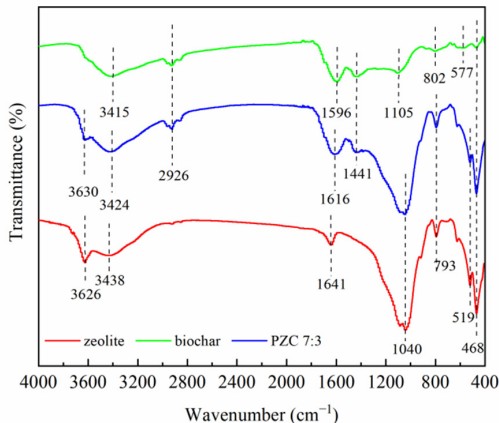

**Figure 3.** FTIR spectra of zeolite, biochar, and PZC 7:3 before phosphate adsorption.

### 3.2. Adsorption Experiments

### 3.2.1. Effects of Dosage on Phosphate Adsorption

The dosage that influenced phosphate adsorption is shown in Figure 4. An increased dosage of biochar and PZC 7:3 enhanced the removal rate of phosphate. In addition, the phosphorus removal rate rapidly increased when the amount of biochar and PZC 7:3 increased from 0.2 to 0.8 g, but when the amount exceeded 0.8 g, the increase in the phosphorus removal rate gradually decreased. The dosage of the adsorbent significantly influenced the adsorption efficiency. It is critical to seek out the optimum dosage of adsorbent for contaminants' removal in order to enhance its cost effectiveness [52,53]. In this experiment, when the dosage was raised from 0.2 to 1.2 g, the removal rate of PZC 7:3 rose from 14.62% to 93.50%. This result showed that the adsorbent provided more adsorption sites due to the increased dosage. Then, phosphate was easier to bind to the adsorption sites, resulting in a greater removal rate. However, when the adsorption reached saturation, the total amount of adsorbed phosphate did not change [52,54]. Therefore, according to the dosage-influenced phosphate removal rate (Figure 4), the dosage of 0.8 g was used for subsequent experiments.

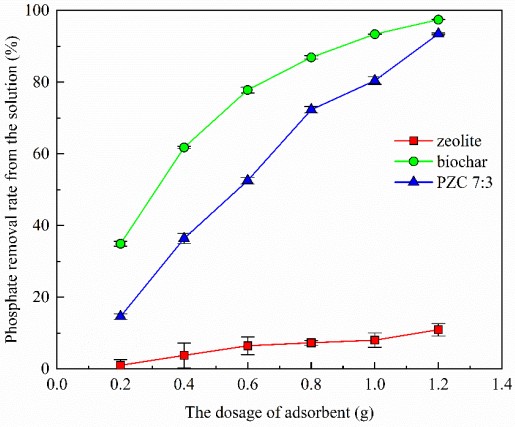

**Figure 4.** The influence of dosage on phosphate removal rate.

### 3.2.2. Influence of the Preparation Method on the Removal of Phosphate from Water

This study used two preparation methods: pyrolysis of zeolite and then mixed with biochar (PZB), and co-pyrolysis of the mixture of corn straw and zeolite (PZC). The results of removing phosphate from the solution with these sorbents are shown in Figure 5a. The results demonstrated that biochar was the best sorbent for removing phosphate from the solution, and zeolite had the lowest adsorption capacity. Furthermore, PZ exhibited an increased ability to remove phosphorus compared with zeolite, indicating that thermal modification can enhance the amount of phosphorus that can be adsorbed [27]. The co-pyrolysis of zeolite and corn straw material removed phosphorus significantly more than the direct mixing of pyrolyzed zeolite with biochar. It was shown that pyrolysis increased the adsorption of phosphorus to zeolite, and the substances released by the carbonization of straw during co-pyrolysis (such as organic acids) could affect the modification of zeolite [55]. However, this view requires further observational study. In addition, the PZC adsorption capacity was equal to 90% of biochar for removing phosphate (Figure 5a).

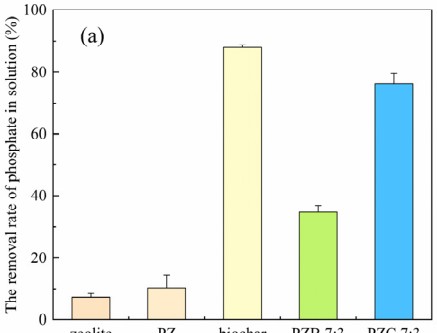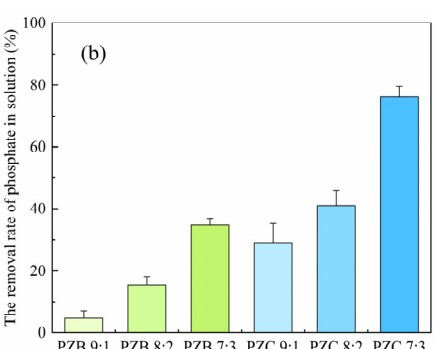

**Figure 5.** The removal rate of phosphate in solution of differential materials. (**a**) Influence of preparation methods on phosphate removal. (**b**) Influence of different ratios (biochar:zeolite) on phosphate removal. Different colors refer to various materials and results in this section.

### 3.2.3. Influence of Different Ratios (Biochar:Zeolite) on the Adsorption

The adsorption of phosphate at different ratios of PZB and PZC is shown in Figure 5b. The capacity of different ratios of PZB and PZC for removing phosphate was increased as the corn straw was increased. It was demonstrated that biochar performs an important function in the adsorption processes of composite materials [56]. When the ratio was 7:3, the removal rate due to the zeolite-biochar composite material (PZC 7:3) was closer (90%) than that of biochar. Therefore, the ideal composite material for absorbing phosphorus in water is the co-pyrolysis of zeolite and biochar feedstock with a ratio of 7:3.

### 3.2.4. Influence of Solution pH on Phosphate Adsorption Capacity

The pH level in the solution is an important factor in determining the adsorption potential of different pollutants [57]. In this work, the effects of different adsorbents on phosphate adsorption at pH values from 3 to 11 were investigated. As displayed in Figure 6, the pH of a solution affected phosphate removal. The adsorption capacity of PZC 7:3 was raised from 0.85 to 1.01 mg/g in the pH range of 3–7, which is contrary to the studies that reported that adsorption remained unchanged or decreased in the pH range of 4–8 [58,59]. In acidic environments, phosphorus mainly exists in $H_2PO_4^-$, and $H_2PO_4^-$ can be adsorbed on PZC by electrostatic attraction or interact with the hydroxyl groups on the surface of PZC via ligand exchange [60,61]. When the pH was between 7 and 11, the adsorption capacity of PZC 7:3 for phosphate was found in the range of 1.01–1.06 mg/g. The calcium in zeolite can precipitate with $HPO_4^{2-}$ or $PO_4^{3-}$ [8].

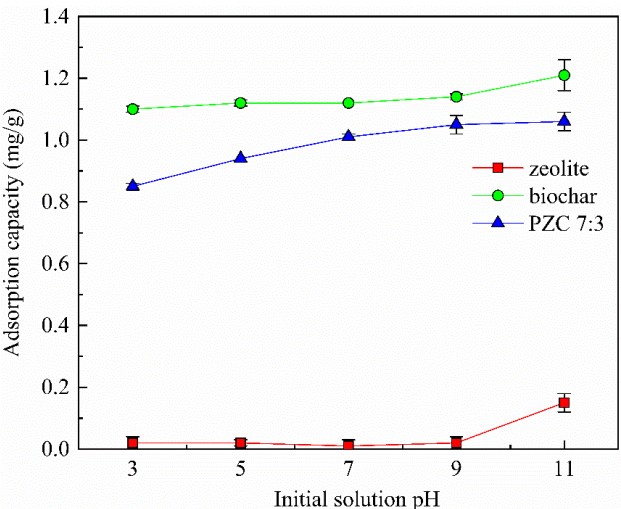

**Figure 6.** The effects of pH values on the adsorption capacity of zeolite, biochar, and PZC 7:3.

### 3.3. Adsorption Kinetics and Isotherm Adsorption

#### 3.3.1. Adsorption Kinetics

As shown in Figure 7, the phosphate adsorption capacity of zeolite, biochar, and PZC 7:3 varied with contact time under constant temperature conditions. In the dynamics of phosphate adsorption, three phases of adsorption, such as rapid, equilibrium, and relatively slow, were displayed. As exhibited in Figure 7, the initial stage of adsorption, which lasted from 0 to 60 min, was called rapid adsorption, and phosphate was adsorbed at 76.88%, 72.05%, and 3.18% by biochar, PZC 7:3, and zeolite, respectively. It was demonstrated that the abundant adsorption sites on the surface of sorbent in the initial stage could interact with phosphate, which resulted in a quick increase in the adsorption capacity [62]. Based on the relationship between the phosphate adsorption capacity and time at this stage, it was hypothesized that zeolite, biochar, and PZC 7:3 mostly adsorbed phosphate on their exterior surfaces by either external surface adsorption or fast boundary layer diffusion [63]. For 60–480 min, the adsorption capacity kept going up because a lot of phosphorus was being absorbed and more of the adsorption sites on the adsorbent were being used [64,65]. After 4 h, the adsorption reached its equilibrium adsorption capacity.

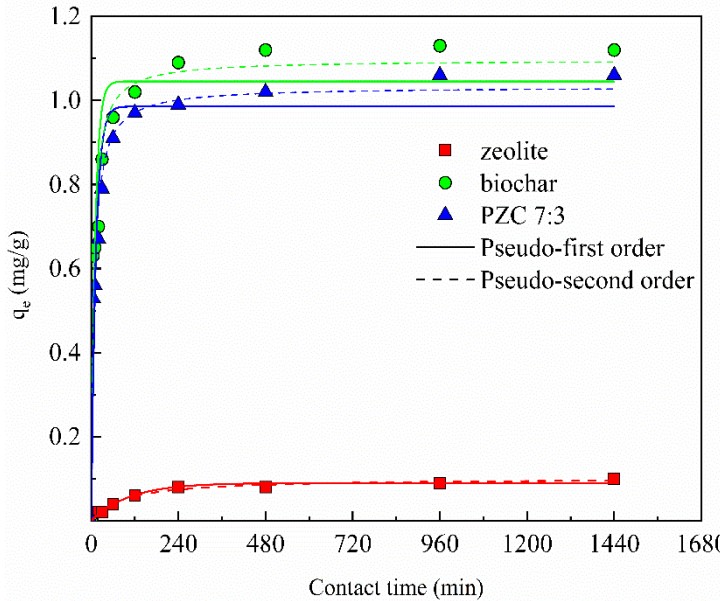

**Figure 7.** Fitting of pseudo-first-order and pseudo-second-order kinetic models.

The adsorption process of phosphate on the adsorbent was investigated with the assistance of pseudo-first-order and pseudo-second-order kinetic models [66]. As represented in Table 3, $R^2$ values of the pseudo-first-order and pseudo-second-order models of zeolite, biochar, and PZC 7:3 were greater than 0.87, showing that the adsorption process was dominated by chemisorption [67], such as ion exchange, precipitation, and ligand exchange [68,69]. In addition, the impacts of physical adsorption cannot be neglected.

**Table 3.** Kinetic parameters of phosphate adsorption.

| Samples | Pseudo-First-Order Model | | | Pseudo-Second-Order Model | | |
|---|---|---|---|---|---|---|
| | $k_1$ | $q_e$ | $R^2$ | $k_2$ | $q_e$ | $R^2$ |
| zeolite | 0.0096 | 0.0901 | 0.9701 | 0.1123 | 0.1016 | 0.9792 |
| biochar | 0.0936 | 1.0452 | 0.8711 | 0.1423 | 1.0965 | 0.9545 |
| PZC 7:3 | 0.0793 | 0.9855 | 0.9102 | 0.1279 | 1.0326 | 0.9734 |

### 3.3.2. Adsorption Isotherm

Figure 8 depicts the adsorption isotherm based on the equilibrium concentration and equilibrium adsorption capacity of the phosphate solution. The equilibrium adsorption capacity of zeolite, biochar, and PZC 7:3 for phosphate was rapidly increased with the increase of the equilibrium concentration at the lower initial phosphate concentration. When starting with a high concentration of phosphate, the equilibrium adsorption capacity grew gradually along with the equilibrium concentration. This phenomenon is consistent with the findings of previous studies [18,54]. This was attributed to the extensive adsorption sites on the surface of zeolite, biochar, and PZC 7:3 at a lower starting solution concentration, which were able to make complete contact with phosphate. Therefore, phosphate was easily adsorbed at a lower concentration by zeolite, biochar, and PZC 7:3. However, in the high concentration, the adsorption sites of zeolite, biochar, and PZC 7:3 gradually reached saturation, and then the adsorption capacity slowly increased to equilibrium.

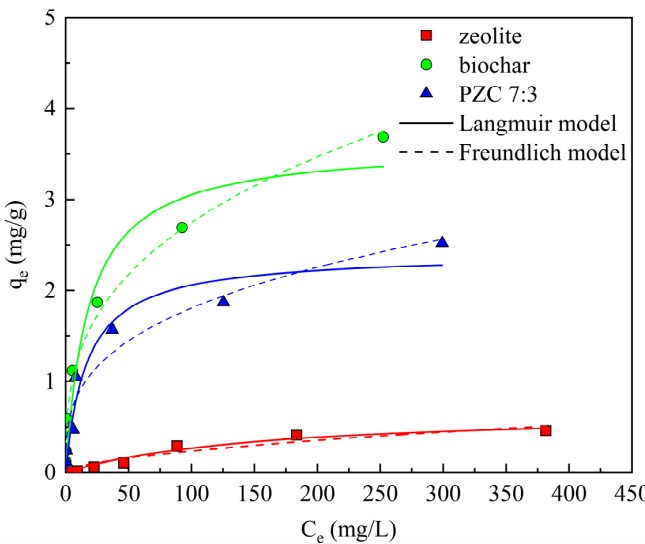

**Figure 8.** Fitting of Langmuir and Freundlich models.

The above experimental findings were used in the Langmuir and Freundlich models to fit the adsorption of phosphate by different materials. The experimental data fitting parameters are shown in Figure 8 and Table 4, where the maximum adsorption capacities of zeolite, biochar, and PZC 7:3 were 0.69, 3.60, and 2.41 mg/g, respectively. Based on $R^2$ (Table 4), the Freundlich model better described the biochar and PZC 7:3 adsorption processes of phosphate, while the Langmuir model better outlined the zeolite adsorption process of phosphate. In the previous publication, it was reported that the Langmuir

equation could be utilized to explain the monolayer adsorption of adsorbates on adsorbents and the Freundlich equation could be used to describe the empirical model of chemisorption of adsorbates on inhomogeneous surfaces [70,71]. It was shown that physical adsorption occurred during phosphate adsorption by zeolite, while chemisorption occurred during phosphate adsorption by biochar and PZC 7:3.

**Table 4.** Isotherm parameters of phosphate adsorption.

| Samples | Langmuir Adsorption Model | | | Freundlich Adsorption Model | | |
|---|---|---|---|---|---|---|
| | $K_L$ | $q_m$ | $R^2$ | $K_F$ | $1/n$ | $R^2$ |
| zeolite | 0.0064 | 0.6874 | 0.9540 | 0.0165 | 0.5775 | 0.8910 |
| biochar | 0.0552 | 3.6048 | 0.9437 | 0.5834 | 0.3366 | 0.9899 |
| PZC 7:3 | 0.0594 | 2.4052 | 0.9364 | 0.4146 | 0.3196 | 0.9430 |

### 3.4. Adsorption Mechanisms and Analysis

In most cases, the mechanism of phosphate adsorption on various adsorbents is mostly attributable to the influences of electrostatic gravitation, ion exchange, ligand exchange, and surface precipitation [72,73]. In this study, the results of adsorption experiments and the characterization of the materials before and after phosphate adsorption were examined in order to identify the adsorption mechanism of zeolite-biocarbon composites for the removal of phosphate from aqueous solutions.

### 3.4.1. Material Microstructural Characterization after Phosphate Adsorption

Compared with the samples before adsorption, FTIR spectra of zeolite, biochar, and PZC 7:3 after phosphate adsorption exhibited significant changes, such as enhancement and shifting of the characteristic peaks and the appearance of new characteristics peaks (Figure 9). This indicates that phosphate reacted with functional groups during the adsorption processes. After phosphate adsorption, all three materials showed a weak characteristic peak near 2320 cm$^{-1}$, which was caused by P-H stretching vibrations. Biochar showed vibration peaks at 1377 and 574 cm$^{-1}$ due to P=O stretching [74]. This demonstrates that phosphate could be absorbed on the surface of the material, most likely through hydrogen bonding or metal bond bridges, etc. Biochar and PZC 7:3 showed a significant shift in the position of the O-H stretching vibration peak at 3200–3600 cm$^{-1}$. This indicates that the functional groups in the material chemically interacted with phosphate, possibly by ligand exchange between O-H and phosphate [43].

Figure 10 depicts the TEM and EDS images of zeolite, biochar, and PZC 7:3 after phosphate adsorption. As depicted in Figure 10A, biochar-adsorbed phosphate formed a sheet-like thin-film substance, and the diffuse ring was observed by the selected area electron diffractogram as characteristic diffraction of the amorphous state. This demonstrates that the thin-film substance is an amorphous mass. As shown in Figure 10a, when biochar adsorbed phosphate, P content increased significantly, proving that P was successfully loaded on the biochar. The contents of Mg, Ca, and P in this substance were positively correlated, which may be an amorphous thin-film-like calcium and magnesium phosphate compound, formed by Ca and Mg oxides in biochar and phosphate (Cu is the analytical background component of the instrument). As represented in Figure 10B, after adsorption of phosphate, zeolite had an uneven and irregular distribution with almost no change in morphology, in which diffraction spots were observed through the electron diffraction pattern of the selected area, indicating that the substance has a crystalline structure. The EDS analysis revealed that the material contained a large amount of O, Si, Al, and Mg (Cu is the analytical background component of the instrument) and a small amount of K, Ca, and P (Figure 10b). In this case, zeolite only adsorbed a small amount of phosphate, and there was no phosphorus-enriched phase, reflecting that the phosphorus may be dispersed as ions on the surface or within the pores of the zeolite. As shown in Figure 10C, some thin-film-like substances were attached to the edge of the zeolite-biochar composite, which is similar to

that displayed in Figure 10A. As shown in Figure 10c, EDS analysis showed a great amount of C, O, Ca, Mg, and P in the substance (Cu is the analytical background component of the instrument). This indicates that the Ca and Mg oxides in the zeolite-biocarbon composite were formed with P as calcium and Mg phosphate compounds and attached to the zeolite surface, demonstrating that the composite of zeolite and biochar improved the adsorption of phosphate.

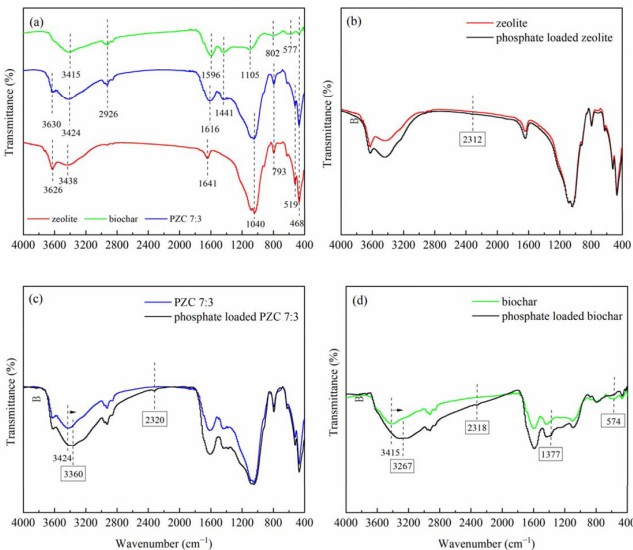

**Figure 9.** FTIR spectra of zeolite, biochar, and PZC 7:3 before and after phosphate adsorption. (**a**) Zeolite, biochar, and PZC 7:3 before adsorption. (**b**) Zeolite before and after adsorption. (**c**) Biochar before and after adsorption. (**d**) PZC 7:3 before and after adsorption.

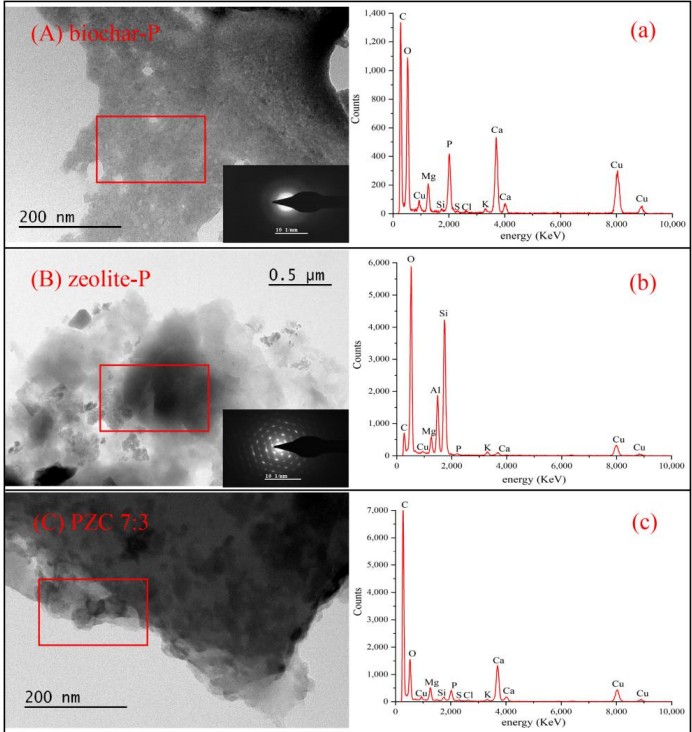

**Figure 10.** The TEM and EDS images of different adsorbents after adsorption of phosphate (TEM images of (**A–C**) representing biochar-P, zeolite-P, and PZC 7:3-P, and EDS images of (**a–c**) representing biochar-P, zeolite-P, and PZC 7:3-P, respectively).

### 3.4.2. Adsorption Mechanisms

The mechanism of phosphate adsorption by the zeolite-biocarbon composites prepared in this study was mainly attributed to ligand exchange, followed by electrostatic gravitational force and an ion exchange effect (Figure 11).

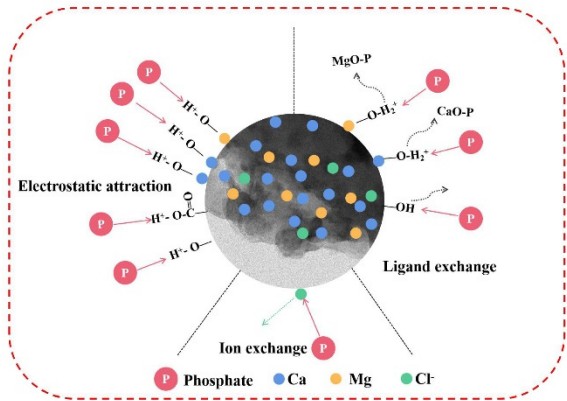

**Figure 11.** Mechanism of phosphate adsorption by zeolite-biochar composite (PZC 7:3).

1.  Electrostatic attraction

Adsorption of phosphate onto the adsorbent was mainly achieved by the attraction of opposite charges by electrostatic induction [71]. Commonly, phosphate in solution exists mainly as $H_2PO_4^-$, $HPO_4^{2-}$, and $PO_4^{3-}$. Zeolite-biochar composites contain a large amount of calcium and magnesium oxides that should be partially protonated in an aqueous solution to be positively charged [25,54]. When the pH of a solution is smaller than the pH of the metal oxides, also known as a zero-point charge, the positively charged zeolite-biochar composite has the capability to absorb the negatively charged phosphate via the process of electrostatic gravity.

$$Ca(Mg)O + H_2O \rightarrow Ca(Mg)\text{-}OH^+ + OH^-$$

$$Ca(Mg)\text{-}OH^+ + H_2PO_4^-/HPO_4^{2-}/PO_4^{3-} \rightarrow Ca(Mg)\text{-}OH^+\text{-}H_2PO_4^-$$

2.  Ligand exchange

It was displayed that the phosphate adsorption process occurs chemically according to the movement of O-H peaks in the FTIR pattern of the zeolite-biochar composite after phosphate adsorption and the generation of amorphous thin-film calcium and magnesium phosphate compounds in TEM-EDS. This is because the surface of the zeolite-biocarbon composite was covered with MgO and CaO particles, which can be used as the primary adsorption sites for removing phosphate. When it comes into interface with water, hydroxylation of the metal oxide surface can occur, and phosphate replaces the hydroxyl group for ligand exchange [75], resulting in the formation of an amorphous calcium and magnesium phosphate compound.

$$Ca(Mg)O\text{-}OH_2^+ + H_2PO_4^- \rightarrow Ca(Mg)O\text{-}H_2PO_4 + H_2O \ (0.12 < pH < 9.21)$$

$$2Ca(Mg)O\text{-}OH_2^+ + HPO_4^{2-} \rightarrow 2(Ca(Mg)O)\text{-}HPO_4 + 2H_2O \ (5.21 < pH < 10.67)$$

$$3Ca(Mg)O\text{-}OH_2^+ + PO_4^{3-} \rightarrow 3(Ca(Mg)O)\text{-}PO_4 + 3H_2O \ (10.67 < pH < 12)$$

In addition, it has been shown that metal oxides can generate surface precipitation with phosphates through hydrogen or chemical bonding [76–78]. Although no solid precipitation was seen for any of the three materials in this study, an amorphous thin-film material was discovered, which may be connected to the lower phosphate content or the shorter aging period of the precipitation [79].

3.　　Ion exchange

The presence of $Cl^-$ in the EDS of zeolite-biocarbon composites after adsorption of phosphate indicates its involvement in the adsorption process, as well as the possibility of ion exchange between phosphate and chloride ions at the adsorption site. Mosa et al. [43] evinced that the higher the $Cl^-$ content in the material, the more actively the ion exchange mechanism is involved. The relatively low chloride ion content in this study suggests that the adsorption process was successful.

*3.5. Desorption Studies*

As shown in Figure 12, the desorption rates of zeolite, biochar, and PZC 7:3 for phosphate were 76.03%, 44.85%, and 20.82%, respectively. The reason was that the adsorption of zeolite on phosphate was mainly physical adsorption with weak intermolecular force bonding, which led to the phosphate being easily desorbed. In contrast, the adsorption of phosphate by PZC 7:3 was mainly chemisorption with strong chemical bonding. Therefore, it was not easy to desorb, indicating that the zeolite-biochar composite was more stable for phosphate adsorption than zeolite and biochar alone. In addition, the adsorption of phosphorus by zeolite-biochar composites could enhance its availability as a slow-release fertilizer in the soil, allowing phosphate to be retained in the soil for a long period of time to supply nutrients to the plants [80,81]. This requires further research.

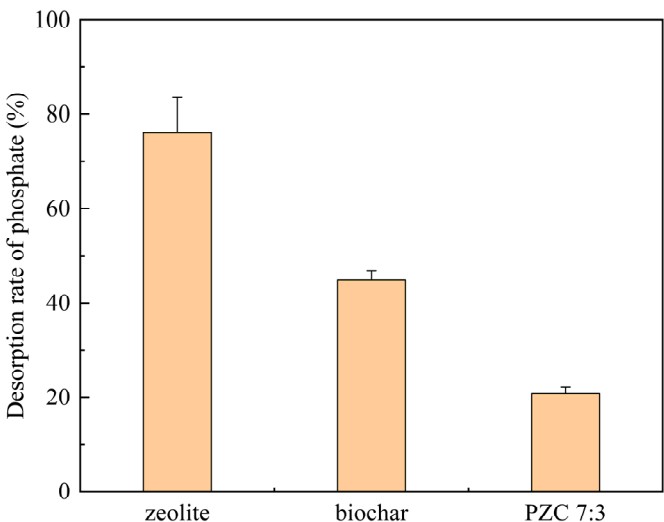

**Figure 12.** Desorption rate of phosphate by zeolite, biochar, and PZC 7:3.

**4. Conclusions**

In this study, the zeolite-biochar composite material was prepared at different ratios to examine the adsorption of phosphate from water. In addition, the physicochemical features of the zeolite-biochar composite have been evaluated, in which the impacts and processes of phosphate adsorption have been investigated. The findings revealed that the optimal phosphate adsorption effect was observed when the biochar composition accounted for 30% and the composite material was prepared with the co-pyrolysis method. The maximum adsorption capacities of zeolite, biochar, and PZC 7:3 were 0.69, 3.60, and 2.41 mg/g. The main mechanism of phosphate removal by PZC 7:3 was the formation of thin-film amorphous calcium-magnesium phosphate compounds by ligand exchange. The adsorption process was also influenced by electrostatic attraction and ion exchange.

In summary, considering the disadvantages of using zeolite and biochar alone, the C/H of the zeolite-biochar composite was lower than that of biochar, indicating that the combination of zeolite enhanced the stability of the composite. The oxygen-containing functional groups of biochar were added during the co-pyrolysis process. Compared to the adsorption of phosphate by zeolite, the adsorption capacity of phosphate by the

zeolite-biochar composite was greatly enhanced. Experiments with desorption showed that the phosphate adsorption of the zeolite-biochar composite was more stable than the phosphate adsorption of either zeolite or biochar by themselves. Therefore, this study indicates that PZC 7:3 could be used to remove phosphate from water because it is a stable and effective adsorbent.

**Author Contributions:** Z.D.: conceptualization, methodology, writing—original draft, data curation; S.G.: conceptualization, writing—review and editing; H.C.: conceptualization, funding acquisition, writing—review and editing; D.X.: resources, methodology; G.T.: improvement of English language; X.W.: methodology, formal analysis, visualization; W.N.: data curation, formal analysis; M.M.: methodology, visualization. All authors have read and agreed to the published version of the manuscript.

**Funding:** This research was funded by the National Key Research and Development Program of China (2018YFC1802601), the "Light of West China" Program, and the Opening Fund of the State Key Laboratory of Environmental Geochemistry (SKLEG 2022216). Above fundings were from Institute of Geochemistry, Chinese Academy of Sciences.

**Institutional Review Board Statement:** Not applicable.

**Informed Consent Statement:** Not applicable.

**Data Availability Statement:** Not applicable.

**Conflicts of Interest:** The authors have no relevant financial or non-financial interests to declare.

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
