# Peer review of "Removal of Phosphate from Aqueous Solution by Zeolite-Biochar Composite: Adsorption Performance and Regulation Mechanism"

_applsci, doi:10.3390/app12115334_

Round 1

Reviewer 1 Report

Reviewer comments:

The research design of the manuscript: “Removal of phosphate from aqueous solution by zeolite-biochar composite: adsorption performance and regulation mechanism” is appropriate, with clearly presented parts: Introduction, Experimental, Results and Discussion and Conclusions at the end. I have only few comments: 

Part 2, Materials and methods, page 3, line 105-106

Can the authors determine the content of each mineral phase (clinoptilolite, montmorillonite and quartz) in the zeolite tuff? What was the cation exchange capacity of the tuff? What was the size of the zeolite sample? 

The authors did not investigate the potential leaching of the phosphorous after the adsorption, so I have the recommendation for them to do that.

Author Response

Response to Reviewer 1 Comments

1. Comment: Part 2, Materials and methods, page 3, line 105-106

Can the authors determine the content of each mineral phase (clinoptilolite, montmorillonite and quartz) in the zeolite tuff? What was the cation exchange capacity of the tuff? What was the size of the zeolite sample?

Response: Thanks to the reviewers for your constructive suggestions. Actually, we determined the content of each mineral phase (clinoptilolite, montmorillonite and quartz). It is our neglect that these results were shown in the first version. According to the comments, I have added this information in the modified version (Line 106-109). Thanks again for this comment.

2. Comment: The authors did not investigate the potential leaching of the phosphorous after the adsorption, so I have the recommendation for them to do that.

Response: Thank you for pointing out this problem in our manuscript. According to the revised content, we supplemented the desorption experiment of phosphate, and investigated the desorption rate of phosphate by zeolite, biochar and PZC 7:3. The desorption experiment part is in line 202-213, the desorption experimental results and discussion section are in line 537-548. In addition, The desorption rate of the phosphate is shown in Figure 12.

Reviewer 2 Report

This manuscript by Deng et al. aims to assess the adsorption performance of a zeolite/biochar composite for phosphorus removal.

This manuscript is correctly written and the experiments have been diligently conducted. The only point that deserves for improvement is the justification of biochar/zeolite composite, although the best adsorption performance was for biochar. This must be better explained, and not only in the introduction. Beyond this minor point, the manuscript can be accepted.

Author Response

Response to Reviewer 2 Comments

1. Comment: This manuscript is correctly written and the experiments have been diligently conducted. The only point that deserves for improvement is the justification of biochar/zeolite composite, although the best adsorption performance was for biochar. This must be better explained, and not only in the introduction. Beyond this minor point, the manuscript can be accepted.

Response: We sincerely thank the reviewer for the careful reviews and the positive remarks on our work! According to your kind and construction comments and suggestions, we added the advantages of zeolite-biochar composite for phosphate adsorption compared to zeolite and biochar alone for phosphate adsorption in the conclusion section of the article (Line 563-571). Thanks again for your objective suggestions.

Round 2

Reviewer 1 Report

The authors of the manuscript entitled “Removal of phosphate from aqueous solution by zeolite-biochar composite: adsorption performance and regulation mechanism” has answered to all questions presented in the first round of the reviewing process and the manuscript is now improved. I recommend this manuscript to be accepted in this form.